# Early Identification of Risk of Child Abuse Fatalities: Possibilities and Limits of Prevention

**DOI:** 10.3390/children9050594

**Published:** 2022-04-22

**Authors:** Ivana Olecká

**Affiliations:** Department of Christian Social Work, Sts Cyril and Methodius Faculty of Theology, Palacký University Olomouc, 779 00 Olomouc, Czech Republic; ivana.olecka@upol.cz

**Keywords:** fatal abuse, fatal neglect, maltreatment, child

## Abstract

(1) Background: The aim of the study was to analyse the structure of registered fatal violent crimes against children under 5 years of age and to identify the main characteristics and risk factors of fatal violence against children in order to discuss the possibilities and limits of prevention of these crimes. (2) Methods: Mixed-method design: 1. retrospective statistical analysis of data extracted from Czech statistics about crime. 2. qualitative analysis of autopsy reports and construction of serial case study. The data were pooled from two different sources: 1. Statistics about crime against children aged 0 to 5 (n = 512). 2. Autopsy reports (n = 52) of children up to the age of five. (3) Results: The following indicators and risk factors were identified: mental disorder or cognitive deficits in parents, parents’ immaturity, poor parenting skills, inadequate parenting practices, absence of a deep emotional bond with the mother, lack of parents’ interest in catering to the children’s needs, parents’ addiction, an unprotected, hazardous environment and surroundings, household falling apart, incidence of suspected domestic violence, incidence of multiple bruises and untreated injuries, aggressively dominant parents, poverty, absence of adequate health care, medical neglect of a child, poor health of the child and failure to thrive. (4) Conclusions: The task for the state is to make effective use of all accessible mechanisms to improve the situation in families. Particularly in the context of the newly emerging situation of increasing uncontrolled violence in families in the context of the restrictions of the COVID-19 pandemic, this demand is more than urgent. Close attention should be paid to children who are not registered with pediatricians and fail to attend regular medical examinations. It is also vital to follow families in which violence has already been suspected in the past.

## 1. Introduction

Although a child fatality caused by maltreatment, neglect, or abuse is a very sensitive and serious social issue, studies tackling extreme forms of child maltreatment, neglect or abuse in families are almost non-existent as are aggregated valid data on violence against children. The necessity for prevention, especially in the under 5 age group, has already been pointed out by the World Health Organization (WHO) in the document Health 21 [1]. The foundations of social and legal protection of children in the Czech Republic are based on the Charter of Fundamental Rights and Freedoms, where the principles for the protection of parents, family, children and adolescents and labor law guarantees pregnant women and adolescents are enshrined in Article 32. The latest consolidated fifth and sixth periodic reports on the implementation of the obligations under the Convention on the Rights of the Child shows that the main strategic document for the implementation of the Convention is the National Strategy for the Protection of the Rights of the Child adopted by the Government in 2012. Nevertheless, works dealing with extreme forms of child abuse, neglect and abuse in a family in this age group are practically absent and represent a rather marginal issue in references on this syndrome [2,3].

Child maltreatment (CM) [4] (physical abuse, sexual abuse, neglect, medical neglect, and psychological maltreatment) is a set of adverse symptoms in various areas of the child’s condition and development as well as their position in society, especially in the family, and is mainly the result of intentional harm to the child, caused or inflicted most often by their closest caregivers, mainly parents. This should not be comprehended as a unilateral act of the actor or perpetrator [5]. Instead, it should be viewed as an odd interaction of all actors concerned and conditions in which such a process is taking place, largely within a framework of social and cultural patterns in relationships between adults and children; parental status and its significance in society; and its legal basis and consequential value of a child in society and naturally, the parents’ personality profile [5,6,7]. According to Dunovsky [5], infanticide is the most pronounced form of CM. Child death due to CM can be considered a separate category of infanticide [8,9,10,11,12,13,14,15,16,17,18].

We learn about risk factors mainly from analyses of crime statistics, mortality records, or data from various NGOs. We can divide the main risk factors into four groups: 1. parent–child interaction; 2. parent characteristics independent of the child; 3. child characteristics, excluding parents; and 4. family factors. According to a meta-analytic review of the literature made by Stith at al. [19], there are 39 different risk factors. The large effect sizes were found between child physical abuse and three risk factors (parent anger/hyper-reactivity, family conflict and family cohesion). Large effect sizes were also found between child neglect and five risk factors (parent–child relationship, parent perceives child as problem, parent’s level of stress, parent anger/hyper-reactivity, and parent self-esteem). Other studies emphasize factors such as younger age [5,20,21,22,23,24], physical health problems and vulnerability [25,26,27,28], race [29], single parent status [22,23,30], parents’ own history of maltreatment [31], substance abuse [20,32,33], high levels of stress [27,34], low levels of social support [27,34], mental health problems [5,31,35], larger numbers of children in the household [25,26,36], domestic violence [34] and low family income [33,35,37]. However, there is no aggregate valid data on fatal violence against children [38,39,40,41,42,43]. This fact has also been analysed in our previous research [44,45]. Infants are most often affected by the fatal consequences of abuse and neglect, while many of these cases remain undetected or unexplained, probably due to the limits of sudden death diagnostics [22,44,46,47,48,49].

Studies [22,50,51,52,53,54] confirm that the most frequent perpetrators are a child’s parents, whereas fathers are more often guilty of their death as a result of fatal abuse and the mothers as a result of fatal neglect. In terms of a fatality resulting from CM, neglect seems to be an even higher risk phenomenon than maltreatment [50,55]. Paradoxically, society fails to pay much attention to this type of inappropriate child treatment.

Death is considered a direct consequence of extreme neglect by Sidebotham et al. [56], Damashek, Nelso, and Bonner [50], Putnam-Hornstein [57], and Sidebotham and Retzer [58]. Younger children are at greater risk of direct violence, blunt instrument use, and shaking compared to older children [12,17,59,60]. Mothers are liable for deaths of younger children and a number of studies have focused on them as the perpetrators [61] and fathers are more frequently responsible for deaths of older children [62,63,64,65]. The third most frequent perpetrator was the mother’s boyfriend/partner [66,67,68,69]. In terms of a social demographic profile, analysed studies led us to identify the following basic risk factors: low socio-economic status poverty, unemployment, and minimal education, hereinafter only referred to as a lower socioeconomical status, mother—a sole breadwinner, infant children and younger mothers [11,22,51,52,57,70]. Alcohol and/or drug abuse [65,71] or a history of criminal activity [72] are also considered to be risk factors. A history of mental health problems also appears to be problematic [14].

Quite often, these cases are not being identified as risky at all [50,52], even though risk identification acts as a preventive factor [51]. In fact, physical abuse and murders found in a parent’s history present a significant risk factor for fatal child abuse [22,57].

The aim of the study was to analyse the structure of registered fatal violent crimes against children under 5 years of age and to identify the main characteristics and risk factors of fatal violence against children.

## 2. Materials and Methods

Knowledge of the structure of crime against children is an essential prerequisite for designing targeted prevention [22,73]. In line with current research trends in this area, a mixed design was used for data collection and analysis [74]. To describe the current status, data were pooled from two different sources:Data extracted from statistics of crime in the Czech Republic (operated by a specialized workgroup within the Presidium of Police, i.e., Department of Material Responsibility and Statistics of the Criminal Police and Investigation Service Bureau of the Czech Police, hereinafter only referred to as CP). The original data are part of and are permanently stored in the official database Evidence Statistical System of Crime—ESSK. The data are classified, and access is under special consideration. The access was granted by official authorities to provide this research. The data relate to criminal offenses where the affected object (victim) was a child aged 0 to 5 years during the monitored period 2010–2019 (n = 512). During this period, all recorded crimes against children under the age of 5 were included in the file for this purpose. The sampling was based on a cross-section of the years 2010, 2014 and 2019. (Submitted data does not include all information pertaining to traffic accidents). The cross-section of years represents a sample from the base population and is considered a representative sample in terms of the period under study. The dataset defined by us can be quantified in the above-mentioned ESSK under tactical-statistical specific codes [75]. According to Act 40/2009 Coll. of the Penal Code (until 01/01/2010 of Act 141/1961 Coll. of the Penal Code), these are mainly crimes defined in Title I—Crimes against life and health, in Title IV—Crimes against family and children and in Title VII—Generally dangerous crimes.Autopsy records (n = 52) of children who died suddenly, unexpectedly and violently up to the age of five; all of these autopsies were conducted at one forensic medicine workplace in the Czech Republic upon suspicion of CM, based on reports. The autopsy files represent an absolute selection from a set of data extracted from the statistical surveys of crime for the aforementioned period from one territorial unit within the Czech Republic, which falls under the jurisdiction of the investigated forensic medicine department. Ten files were then selected from this set for a qualitative serial case study For the selection the main criterium was the extent of information on the social history of the case. Only cases with the most complex and richest social history data were selected. The same design of analysis was applied to all cases according to Yin recommendation [76]: the analysis of the first cases served as basis for a new theory and further cases served as a verification group. Then the mutual comparison of individual cases served as a new basis. All ten selected cases were analysed in this way and were consequently added to the analysis. The final basis served as a source for research results. The procedure in the event of death and the requirements related to performing an autopsy are set out in Act 372/2011 Coll. on Health Services. According to this law, children are subject to a compulsory autopsy.

On the basis of the selected research design, the collected data were analysed independently through two different methods.

The first was a retrospective statistical description of data extracted from crime statistics: after data cleaning and removal of duplicates, the extracted data were processed through a secondary statistical analysis. Due to the nature of the data, this was an analysis of socio-demographic variables. The following variables were monitored: age and sex of the perpetrator and victim, type of attack, and consequences of the crime.In parallel with the above data analysis, data from 52 autopsy reports were analysed qualitatively. All the cases (n = 52) were categorized via a thematic analysis and were then mutually compared. The thematic analysis focused on identifying and describing implicit and explicit ideas in data, through the identification, analysis and interpretation of meaning patterns (“themes”). Codes were created to represent the identified themes and were applied and were linked to the raw data as aggregate data. The analysis itself involved the following: a comparison of the frequencies of the codes and identification of code co-occurrence. Based on this analysis, risk factors for fatal child abuse were identified. The identified patterns across the population were validated and analysed in more detail through serial case studies. The categories for analysis were also identified through thematic analysis. A non-structured data collection method was used to gather information (units of analysis) from witness statements, police records, photo documentation and other accompanying documents that could be indicators of social risk factors on the part of the mothers of the deceased children. This analysis is conceptualized as descriptive, as it is unfortunate to note that the information was not deep enough in the writings for an interpretive analysis. Subsequently, ten files were selected for the construction of a serial case study. From the set of all available autopsy protocols, a group of cases with rich social history documentation was extracted to allow a more in-depth analysis. The process of data collection started by reading the entire autopsy protocol, while no parts were assessed before the researchers had read the entire text. When comparing the cases based on multiple case study methodology with an embedded approach [76,77], several units of analysis were monitored. Significant emphasis was placed on those units of analysis which in some way pointed to risk factors, warning signs and the connection between the death of a child and CM. The aim of this part of the qualitative analysis was to construct basic categories of risk factors for fatal child abuse. Each file was viewed as a specific case study composed of a descriptive part and an exploration. The descriptive part contained results of medical examination and autopsy of the dead child. Within the context of data contained in the descriptive part, the exploration (qualitative description) tried to reveal risk factors, warning signs and relationships between the child’s death and CM.

## 3. Ethical Aspects and Limits of Research

Through an assessment of data extracted from crime statistics we had to consider that the reported data provided information only about crimes recorded by the Czech police. It can reasonably be assumed that some incidents may show signs of a certain degree of latency. They may go unreported or untracked and their real number may actually be higher.

No research participants were directly contacted as part of the research and the study relied solely on documents. Nevertheless, the analysed material can be considered highly sensitive and the collected data are considered sensitive personal data. Therefore, it was necessary to make sure that none of the involved individuals or institutions in the research could be identified based on the study results. The risk was high, especially in the serial case study. It was decided to remove identification data for the purpose of analysis and the children received fictitious names. Extracts obtained for data analysis is kept in a secure place and will be shredded after completion of research projects that are still ongoing. To inspect the autopsy records, the researcher obtained official consent from the head of the institute in which the autopsy data were collected.

Data extracted from statistics of crime were processed after prior approval of the Department of Material Responsibility and Statistics of the Criminal Police and Investigation Service Bureau of the Czech Police.

Since it was not feasible to combine the research technique of analysis of autopsy records with other methods of data collection—especially questionnaires or interviews with families in which the death occurred—the research suffers from a higher degree of missing information. In many cases, information on social demographic data were missing along with information on the family’s social background. An incomplete Declaration on the Examination of the Deceased and the absence of many documents (where the medical examiner only borrowed documents from the Czech Police or medical records, but did not obtain copies) that could contain this kind of information presents a significant limit on this study that affected the potential of statistical analysis. Any secondary data analysis faces this type of limitation as the data are primarily acquired for a purpose other than the objective of the secondary research. It should also be mentioned with regret that the nature of the data (its structure) did not allow for a deeper statistical analysis to prove causality.

## 4. Results

During the monitored period of the years 2010, 2014 and 2019, the Czech Police registered 512 criminal offences against children up to the age of five. Due to alterations in methodology, the number of criminal offences grew from 135 in 2010 and 133 in 2014 to 244 in 2019. Criminal offences were committed by 409 known perpetrators and of these, 318 were men and 91 were women. It should be stressed that some offences were committed by 2 perpetrators, while 1 perpetrator could have committed more than one offence (e.g., harming more than 1 child), and the file also includes offences with no known perpetrator. There were 221 boys and 291 girls among the victims. The age of the children you can see in Table 1.

The most frequent victims of attacks were children aged four and five. On the other hand, the smallest number of victims were in the category of infants. It is reasonable to assume that this is mainly due to the high portion of latent crime against infants, since our previous findings showed [44] that proving violence against infant children is rather difficult. The most frequent crime was maltreatment (n = 173) and sexual abuse (n = 116), followed by rape (n = 79), intentional injury (n = 50), murder (n = 24), child abandonment (n = 24) and death by negligence (n = 17). Other types of crime accounted for less than 3 percent of all records. Cases where children died as a result of homicide (n = 24), negligent homicide (n = 17) and intentional injury (n = 2) were identified as fatal consequences of the crime.

The data are burdened with a large degree of uncertainty, as almost half of the cases lacked information on the consequences of the crime. In 2010 and 2014, the data on the consequences of crime were not collected and fatal consequences are only inferred from crimes where death as a consequence is evident (e.g., completed homicide). Fatal consequences were suspected based on a crime only where death is evidently a consequence (e.g., a completed murder). Death was proven to have occurred in 8.4% of cases.

The perpetrators of fatal violence are those with a younger average age (ø 23.675) than those who commit violence without fatal consequences (ø 27.849) or those whose crime caused no consequences (ø 37.833). In terms of the age of the victim, children aged one year old are the most likely (n = 22) to die. On the other hand, no case of death of a five-year-old child was recorded in the population. Victims of fatal crime are younger (the average age was 1.488) than victims of crime with non-fatal consequences. These children died as a consequence of murder (n = 24), negligence (n = 17) and intentional injury (n = 2). The ratio between boys and girls—as victims of fatal violence—is balanced (21:22). Compared to other victims of non-fatal crimes, victims of fatal crimes are on average younger. Their average age (mean age) is 1.488. The average age of non-fatal victims is 3.564.

In terms of the relationship with the crime perpetrator, in the case of fatalities, most offenders were parents, while the mother was explicitly stated in only one case. A closer look at the data shows that women-mothers prevail also in the indicator “biological child” (n = 12). Consequently, they form a category of the most probable perpetrators of fatal violence against children up to the age of five.

Despite any reservations about the limitations of collecting data from autopsy protocols, this research provides valuable and interesting initial information about the individual symptoms and risk factors for fatal child abuse and neglect. However, it should be noted that data from autopsy records, which served as a source of information on fatal forms, typically lacked any kind of data depicting the mental state of the children and its manifestations.

Evidence of physical symptoms implies that the most typical signs in fatal outcomes were recurring multiple bruises with unusual localization, the occurrence of untreated injuries and bruises, minor injuries, fingerprints, or tooth prints. In protocols, the presence of these signs usually infers abuse, especially if it is not clear how old these signs are.

As for the risk factors pertaining to physical abuse, approximately half of the children who were victims of fatalities lacked a deeper emotional bond with their mothers (the mother admitted this during the hearing), had parents who were addicts or there was suspicion of violence in the family and an aggressive dominant father/mother in the family. It should also be noted that in ten cases a risk factor of mental disorder or cognitive deficit was identified on the part of the parents.

In the case of negligence leading to fatalities, the following risk factors were identified: a family living on the edge of poverty or in poverty; addiction of parents; poor parental competencies; an insecure, threatening environment in the home and its surroundings; lack of parents’ interest in taking care of to the children’s needs; a household falling apart; or absence of adequate health care. It should also be noted that evidence of malnutrition, i.e., a symptom that could lead to starvation, was identified in only one case of a neglected child fatality and the rest of the children were typically normotrophic. Apart from these factors, the non-fatal cases typically involved long-term unemployment of parents; an immature parent, lack of social support, housing instability, violence within a family and absence of the child’s own space for toys and personal stuff.

A more detailed insight into individual cases is offered in the serial case study of the fatal consequences of inappropriate treatment of children. Based on the results of the analysis, the data are presented in a table. Table 2 shows the sociodemographic characteristics of the children. The series included two infants, two one-year-old children, three two-year-old children, one three-year-old child and one five-year-old child. The gender of the children is balanced 5:5. In all cases, the children had their own mother and in three cases a stepfather. The perpetrator of the fatal violence is highlighted in bold. In five cases, it was the mother, in four cases the father/stepfather and in one case the brother.

The columns of the table are made up of categories that emerged during the analysis. Severe brain swelling after a blunt head injury was identified as the leading cause of death in most cases. In the presented cases, this swelling usually occurred as a result of beating the child on the head with hands (fist or palm). Swelling can also occur as a result of an injury to the skull due to the child’s falling from a height and hitting a hard surface (the aggressor throwing a child on the floor, against a wall or on a bed) or as a result of direct head blows against a hard surface (usually a floor or wall). Unusual blows to the victim’s abdomen leading to peritonitis are not uncommon either. In one case, a child died of suffocation due to strangulation and subsequent failure to take proper care of a child in shock. One case was diagnosed as sudden death syndrome (according to the pathologist, the death occurred without a causal relationship to the changes resulting from accidental injury caused by repeated direct exposure to small-scale blunt violence—insidious beating with an open palm and knuckles of a clenched fist). The predominant cause of death was thus violence of great intensity directed against the child’s head.

During the autopsy, signs of violence (hematomas, wounds, abrasions, fractures, bite marks, fingerprints, etc.) of various dates were found in the children in almost all cases. Long-term physical abuse was inferred from these findings. These signs of violence were found on various parts of the body, including places typical of abuse (behind the auricle, above the scapulae of the hip, in the hairy part of the head, in the face and torso, on the temporal bone, on the neck, on the underside of the penis, on the buttocks). The triggering mechanism of the fatal violence was the child crying or discharge of aggression accumulated during quarrels with a partner or “disobedience” by the child.

Warning signs pointing out the risk of abuse vary from one case to another; however, a common denominator in most cases was ignorance of warning signs by the surrounding community (second parent, neighbors, doctors, nurses, etc.) Even if there was suspected maltreatment or neglect, no one did anything to stop or minimize such treatment of a child. This phenomenon has been interpreted as indifference within the social environment, which is highly tolerant of inadequate parenting methods and considers violence behind the closed doors of households to be a private family matter. However, in the presented cases, the risk was not identified even by professionals (general practitioners, social workers). Inaction by these formal components and the absence of timely intervention contributed to the fatal consequences of long-term abuse.

Low SES is symptomatic for cases of family abuse and neglect. According to our findings, the children were living in inadequate homes (constant dirt and mess present in the household, complete ignorance of hygiene required for proper childcare), they were insufficiently and inadequately fed, their families often experienced quarrels or aggression among partners. Due to the absence of basic care, the children’s autopsies showed signs of failure to thrive (emaciation, low weight, neglected appearance, general decrepitation, bruising at various ages in atypical places, etc.) However, these signs often could not be detected by a general practitioner because in these cases parents ignore regular medical check-ups and even in case of injury resulting from abuse, they fail to seek timely medical assistance for their child. Increased attention should be paid to the absence of regular medical check-ups (especially for families at risk of low SES or for families with a history of violence against children or other members of the household).

Apart from low SES, the presented cases show signs of absence of medical check-ups and the presence of aggressive behavior in the family, along with a whole range of other risk factors of CM. For instance, a child’s poor health (physical and mental disability), which increases the demands on childcare, leads to inadequate responses of parents to the stress induced by this situation. In the presented cases, parents view care for a child with a disability as extremely demanding and emotionally unsatisfactory. Lacking an emotional bond between the mother and the child also plays an important role. This risk factor appears to be significant, even in cases of children without health complications, both for abuse and neglect (typically a combination of them). The absence of a mother’s attachment to a child is often enhanced by prolonged hospitalization (in case of children with disability) or stays in an infant care center (in case of unwanted children and in families with low socio-economic status). Mothers stated that after a long separation from a child they could not get into the habit of taking care of the child and said they “don’t like them as much as they used to”. Notable in these cases is a significant improvement in the health of the child during hospitalization/placement of the child in institutional care and its subsequent deterioration after returning to the family.

In cases of extreme and frequent violence against children, parental immaturity is especially evident (apart from the obvious aggressive nature of the parent). These parents opt for inadequate parenting practices (especially heavy corporal punishment: beating black and blue, cold showers, forcing to kneel or stand for a long time, wrapping in a blanket to prevent access to oxygen), the intensity of which increases over time and consequently leads to the death of the child.

In their testimonies, parents describe their children as disobedient, weepy and clumsy. Their parenting practices were intended to improve the situation, whereas in reality, they were causing their children deep physical and psychological trauma. To list just a few examples: 1. forcing a child to stand for a long time to make their legs stronger and start to walk (the standing is forced by beating); 2. an attempt to put the child to sleep by shouting (the child’s cry is forced by wrapping in a blanket and beating over the duvet until the child falls asleep); 3. toilet training (blows to the body as punishment for failure to hold stools, cold showers at the nape of the neck, a boy had to kneel with his buttocks raised and arms outstretched, into which father put objects, kicked the child, beat him on the fingers with an axe handle or fly swatter, wrapped a rope around his ankle and hung him head down with his face covered).

The presented serial cases aptly illustrate the picture of the fatal form of CM. Individual cases have many common features. We consider institutional indifference and inattention from the social environment to be the essential problem. Ignorance of the warning signs and a high degree of tolerance for domestic violence eventually led to the death of the children in the presented cases.

## 5. Discussion

The presentation of the research results follows the logic of progressively deeper information and an increasingly detailed look at concrete and specific data. Characteristics and risk factors that arise from a comprehensive approach to these data (both quantitative and qualitative data) are selected for discussion.

Based on the records of violence against children under 5 years of age, it can be concluded that although violent behavior towards children is more typical for men [62,63,64,65,78,79,80,81], the perpetrators of fatal crimes are in our set in accordance with Corby [61], mainly women-mothers. This fact is undoubtedly related to the structure of violence with fatal and non-fatal consequences where the lower age of the victim increases the chances of fatal consequences. Our previous findings already point to this phenomenon [2,44,82] when demonstrating that mothers are the most common perpetrators of fatal violence against infants. As a child ages, the probability that the perpetrator will be a man (father, stepfather, or mother’s partner) increases. Other research has arrived at the same conclusions [51,83]. This gender aspect is not surprising, considering that in our social and cultural milieu, mothers are the primary caregivers of infants [84]. The nature of demands that such care entails may create stressful situations which, in the case of accumulated risk factors, tend to trigger aggressive behavior [22,44,85]. On the other hand, numerous studies show that mothers and fathers differ in the ways that they tend to kill their children [22,46,86]. While mothers more often choose suffocation (hands, parts of clothing, breasts), fathers choose methods associated with higher physical violence against the child (shaken baby syndrome, fatal aggression, stab wounds and gunshot wounds).

A widely discussed topic is the age of the perpetrator. Studies from abroad as well as domestic studies [5,22,23,24] consider a mother’s young age to be one of the significant risk factors. According to these findings, the young age is associated with immaturity of the parent, increasing the chances of unwanted pregnancy, can end with fatal consequences for the child. However, our previous findings did not confirm this assumption [2,44]. According to the analysis of the autopsy records, the age of mothers-perpetrators correlates with the overall trend of rising average age of first-time mothers in the Czech Republic. On the other hand, it should be noted that compared to the age of offenders causing crime with non-fatal consequences, it seems that the age of offenders causing crime with fatal consequences is still lower. However, the age structure of these victims and the number of siblings (older children with older siblings have older parents) probably play an important role. Due to the lack of data on the birthdates of offenders in primary sources (autopsy files, police statistics), a detailed analysis of offenders’ age structure was not part of this study. Therefore, a hypothesis regarding the influence of the offender’s age on the consequence of a fatality still needs to be verified.

In the crime statistics, physical abuse prevails together with sexual abuse, whereas the crime of neglect is statistically almost irrelevant. However, such data contradict the statistics of the Ministry of Labor and Social Affairs, presenting neglect as the most frequent cause. The same discrepancy can be found in cases of fatal consequences and data concerning the causes of death. In forensic practice (records entered by the Institute for Health Information and Statistics) it is difficult to diagnose manifestations that are not of a physical nature. According to our findings [45], neglect can therefore be inferred rather indirectly in the event of the sudden death of a child. Many of these acts thus remain undetected or unexplained due to the limits of the diagnosis of sudden deaths [48]. As a result of insufficient investigation of the circumstances of deaths and lack of knowledge about a child’s medical history accompanied by diagnostic non-specificity of pathological findings, unnatural causes of the death of children (and especially infants) may remain (and often remain) undetected. There is no single registration system, and thus it is not surprising to find differences in statistics collected by various departments. We believe that the absence of a single system is a serious limitation, hindering the establishment of preventive measures needed for child protection. We presume that fundamental knowledge of the scope of the issue forms an inevitable prerequisite for developing efficient safety nets. At the same time, we are fully aware of the pitfalls that still prevent the collection of complete and valid data. Like Mydlíková [7], we are also fully aware that the capacity (limitations) to identify risk cases poses a fundamental obstacle to the collection of valid data. At the same time, a thorough mapping of the scope of the issue could support efforts to reduce the trivialization of the problem among the general public. Consequently, the entire topic would cease to be taboo and the community around abusers and professionals looking after children’s health and wellbeing would probably be more inclined to notify respective authorities about the suspicion of abuse, which could prevent some cases of fatal consequences resulting from long-lasting domestic violence.

The key source of information on the incidence of crime committed against children is data extracted from crime statistics. However, such data, due to their nature, are limited, especially in terms of completeness. For example, it would be desirable to have an overview of the number of proven crimes against children. Unfortunately, there are no such statistics. Various non-governmental organizations collect biological data (e.g., Safety Line in the Czech Republic) that, however, may not be used as nation-wide statistics. On the other hand, the Institute for Criminology and Social Prevention [87] collects valuable data as part of its research.

In order to enforce efficient primary prevention, it is necessary to maintain valid documentation and to provide incentives for responsible authorities to engage in interdisciplinary cooperation. According to Ševčík [88], it is very advisable to set up interdisciplinary teams of staff from police, departments of social and legal protection of children, social service providers, health care providers and other state, regional and local administration bodies. Importantly, their cooperation will also involve influencing policy development and implementation at various levels, i.e., policy advocacy [89].

Analysis of data helped identify a whole range of risk factors leading to fatal consequences due to inadequate treatment of children. A list of such factors can be found at the end of the study. At this point, for the purpose of discussion, we have selected those that we consider to be the most important ones and at the same time ones that we assume have the greatest potential for success in setting prevention programs. The selection has been made with a full awareness of the fact that an etiology of actions against the life and health of the child can be very broad and ambiguous, or rather multifactorial.

Low SES (poverty, unemployment, and minimum education) of the family presented a significant risk factor in our sample group. Most foreign studies reach the same conclusion [8,11]. Among others, a hypothesis considering low SES as a risk factor relies on the results of numerous epidemiological studies documenting that mortality within a social class rises with a worsened social economic status of the group within the social structure of the state, with lower levels of education, lower levels of income and a growing number of risk factors in their behavior [90,91]. Wilkinson and Marmot [92] also rank low SES among the ten most significant social determinants of health status. Similarly, within the concept of fundamental causes of illness [93], social status plays a key role in one’s health status. Moreover, low SES forms one of the significant risk factors even in cases of death resulting from SIDS [22,86,94]. Inadequate housing conditions (highly neglected household hygiene, an unmaintained household, living in asylum housing or homelessness) can be attributed as a related factor. In addition, low SES and poor housing conditions may be associated with a degree of social exclusion that increases the risk of indifference of the community and reduces the chances of early risk identification. Single women are more often at risk of social exclusion associated with low SES [95].

Another important factor to be noted is a mental disorder or cognitive deficit on the part of the parent. Vargová [7] points out that the ability of parents to take care of a child is also affected by mild or temporary symptoms of psychological issues that may occur as a result of stressful life situations. Typically, the risk rises with the increasing severity of mental health problems. The association of a child’s death with maternal depression is documented by Sanderson et al. [28] Although the causality of this phenomenon is not entirely clear, and there are a number of hypotheses, such as that depressed mothers may be more likely to be smokers (a SIDS risk factor) or that depressed mothers are more likely to hurt their children physically or pay less attention to them. However, the authors of the study do not unequivocally confirm any of these hypotheses. The inability of parents to take adequate care of their child can be caused by many factors. In our research sample, we have mainly identified mental disorder, young age associated with the immaturity of the parent and low (primary) education of the parents. In effect, extreme immaturity or a significantly reduced intellectual ability may lead to the inability of a parent to acknowledge and distinguish the basic needs of the child [7]. The abovementioned factors are closely related to reduced health literacy. Our older research can be used for comparison [44]. Low health literacy negatively affects the comprehension of a child’s diagnostic and treatment process [96] and thus also decisions concerning one’s health [97]. According to our previous findings and studies from abroad [45,50], in cases of negligence or of a similar nature, a complete absence or lack of child supervision played a crucial role. In cases of direct fault, mothers were mainly not familiar with the health care system and could not apply any information into practice (babybox, the possibility of secret birth, assistance from the Department of Social Legal Protection of Children and the use of rescue social networks). Current research data in the presented study revolves mainly around the problem of the absence of medical care, an underestimation of the severity of the situation and failure to seek timely help in an effort to cover up traces of abuse.

Our previous research among mothers of healthy children and interviews with experts clearly show that the basis of quality child care lies in love for the child, which is directly linked to the necessity of developing basic sensitivity and responsiveness towards the child’s body and soul as well as the ability to correctly estimate the situation [85]. Mothers without an interest in the child lack these abilities and their sensitivity to risky situations is thus significantly reduced. As Valúchová and Dobríková [98] show, children with a poor attachment to their mothers are not only more likely to suffer from higher morbidity, but they are also more prone to impaired social, psychological and neurobiological functions. This disorganized attachment is typical, especially for families where parents are drug users. An insecure type of attachment has also been identified in our sample group. It turned out to be very problematic in cases where the child’s poor health increased the demands on childcare and caused inadequate reactions by the parents to the stress caused by this situation. The statements of the mothers indicate that the absence of emotional bonding was even more potentiated by a long separation from the child (long hospitalization, placement in an infant institution). This finding is in line with Nakonečný [99] who states that breastfeeding stimulates the secretion of prolactin, and thus maternal love is regulated by hormones to a certain extent. A mother’s separation from an infant can therefore lower the intensity of maternal instincts. This rationale is completely rejected by gender theories, which, according to Badinter [100], assume that maternal love is a social construct. Contemporary mothers are burdened with very high expectations, which might cause stress if they feel that they are unable to meet these demands [85].

According to our findings, addiction presence is another significant risk factor in child fatalities. Mydlíková [101] points out that the risk increases even if only one family member is an addict but a mother’s addiction carries the highest risk. Problematic use of addictive substances, combined with providing childcare, is a socio-pathological phenomenon reducing the quality of such care. A possible causal nexus of substance abuse with crime (in our case especially violence against children) has long been a widely discussed topic and relevant issue. It is estimated that this context has played a role in 21 to 54 percent of cases of children with CM [102,103]. Nevertheless, there are relatively few studies that focus on the link between substance abuse and child maltreatment, abuse and neglect. The link between the death of a child and the substance abuse of caregivers is difficult to prove unequivocally since it is very hard to document substance abuse and provide a criminally relevant finding that the substance abuse has adversely affected the offender’s psyche or control and cognitive abilities or social behavior to such an extent that they committed a crime as a result of this condition. A connection is subject to many variables (identification of a suspect or accused person as an addictive substance user and the recording of this fact, collection of bio-material (traces) and provision of a forensic expert’s opinion, recording of the results of measures in the registration of criminal proceedings, etc.) [104].

Similarly, previous occurrences of suspicion of family violence are an important factor. Many authors [88] point out an inheritable nature of the risk of violent behavior. Most alarming are cases documenting that a child was left in the care of a parent who had already caused the death of a sibling [45]. We believe that such families deserve due attention. Along with Vildová [105], we strongly believe that in cases where parents were convicted of a violent crime against a child, the capability to care for and raise other children should be automatically assessed and that these children should be included in a group targeted by social and legal protection. Likewise, increased attention should be paid to families where the violent behavior of one of their members has been recorded in the past. In her study, Lemrová et al. [82] pointed out the risk of underestimating such cases, where social networks (formal and informal) that should have protected a child from violence failed completely. She describes the insufficiency of cooperation between general practitioners, the Department of Social and Legal Protection of Children and the police as a fundamental problem. Jelen et al. [106] makes a point of stressing the absence of cooperation by physicians and their unwillingness to report their suspicions to the Department of Social and Legal Protection of Children. Based on our findings, we believe that it is tardy responses from this government department that hinders the process. Everybody involved treats the issue solely from their perspective and their willingness to cooperate remains very low.

In accordance with the findings of Vildová [105], it should be noted that so far no network of social services has been established to help enhance parental skills. Factors such as a mental disorder or a cognitive deficit of a parent, parental immaturity, poor parental skills, inadequate educational practices, absence of a deeper emotional bond with the mother or a lack of parents’ interest to provide for the child’s needs should be minimized via timely training as part of prevention. Training in parental skills is currently lacking, especially for the generation of parents who largely grew up as the only children in the family and because of this they have no experience with childcare. The problem of poor parental skills has already been identified in our previous research of mothers’ health literacy [2,85]. It emphasizes the importance of enhancing parental skills in the area of detecting children’s needs. Parenting as a value should already be conveyed to pupils at school. This recommendation complies with the strategic paper entitled Strategies for Preventing Child Maltreatment by Developmental Stage and Level of Intervention [107]. Lane [108] advises making use of existing tools for risk identification (e.g., SPARK—Structured Problem Analysis of Raising Kids or SEEK—A Safe Environment for Every Kid). Transformation of the current social climate towards being less tolerant to violence against children should form an inherent part of educational efforts.

The task for the state is to make effective use of all accessible mechanisms to improve the situation in families, where removal of a child from a family should also be considered in extreme cases. Particularly in the context of the newly emerging situation of increasing uncontrolled violence in families in the context of the restrictions of the COVID-19 pandemic, this demand is more than urgent [109]. Evidence that prevention is crucial in this regard has long existed [73]. In all these efforts, the interests and welfare of the child should be of primary importance. Close attention should be paid to children who are not registered with pediatricians and fail to attend regular medical examinations. It is also vital to follow families in which violence has already been suspected in the past. Along with Vildová [105], we strongly believe that in cases where parents were lawfully convicted of a violent crime against a child that their capability to care for and raise other children should be automatically assessed.

## 6. Conclusions

According to the crime statistics, violence against children under the age of five is committed three times more often by men than by women. However, the perpetrators of fatal crimes are mostly women-mothers. Crime related to physical abuse dominate the records, followed by acts of sexual abuse, while acts of neglect go almost unrecorded. Victims of crime against children under five years of age are mainly older children (4 and 5 years old); however, infants up to one and a half years are most likely to die as a consequence of such a crime. Perpetrators of such crimes are generally younger too (mode 23.7).

In cases of the fatal form of abuse and neglect, physical abuse prevails, which tends to mainly affect infants. The following is a list of signs and risk factors of inappropriate treatment of children with fatal consequences that have been identified in our sample group (n = 52). It should be noted that these risk factors create a cumulative effect that leads to fatal child abuse in the cases analysed:mental disorder or cognitive deficit on the part of the parentparental immaturitypoor parenting skillsinadequate educational practicesabsence of a deep emotional bond with the motherlack of parents’ interest to provide for their children’s needsa parent or parents’ addictionunprotected, threatening home environment and surroundingshousehold deteriorationthe occurrence of suspected domestic violenceoccurrence of multiple bruises of various ages on odd placesoccurrence of untreated injuries and bruises, minor injuries, fingerprints, bite marksaggressively dominant father/mother in the familya family living on the edge of poverty or in povertyabsence of adequate health carea child does not visit a doctorpoor health of the childsigns of failure to thrive

We believe that the indifference of the surrounding community (second parent, neighbors, physicians, etc.) to such signals poses a great risk. We presume that children’s deaths could be prevented only if social rescue networks would work effectively and risk factors directly leading to fatalities would be recognized in a timely manner. Children themselves rarely ask for help, and if they do, it usually happens only when violent attacks become unbearable. Adults living in the child’s vicinity, but mainly professionals, should be able to notice such adverse events and notify the authorities much sooner.

Dysfunctional social rescue networks (institutional indifference) combined with a lack of interest from community members form the very core of the problem are some of the issues arising from the case studies presented in this study. Neglect of warning signals from the surrounding community (second parent, neighbors, physicians, etc.) is, therefore, accompanied by an unwillingness to share information. We presume that the problem might revolve around a low awareness from the lay and professional public about the extent and severity of the issue. Arguably, the problem may also derive from the above-discussed limited possibility of the responsible authorities to identify risks without the existence of a valid and reliable tool. The presented data offer a very limited amount of information. Underestimation of data and the extent of the problem in our society leads to a trivialization of symptoms and reduces the chance of early risk identification. Likewise, we believe that the absence of a central case registry is a major obstacle. Since there is no legal obligation to report a new place of residence if a family moves to a new address (e.g., cases of asylum tourism), the family will “drop out” of the system. In light of aforementioned, it would be desirable to have more effective coordination through a multidisciplinary team.

We consider the prevention of inappropriate treatment of children to be a cornerstone in the fight against the fatal consequences of the CM. In this context, it is highly desirable to develop an appropriate instrument to make it more effective at capturing the risks of fatal violence against children.

## Figures and Tables

**Table 1 children-09-00594-t001:** Age of the victim.

Age of the Victim	Absolute Frequency	Relative Frequency
0	12	2.3
1	89	17.4
2	64	12.5
3	98	19.1
4	130	25.4
5	119	23.2
Total	512	100.0

**Table 2 children-09-00594-t002:** Serial case study—social demographic data.

Case	Age	Sex	Cause of Death	Parents	Maltreatment/Neglect	Warning Signals	Social History
A	18 months	female	the child was harshly thrown on the floor by the mother	biological mother, age 22, primary education, worker biological father, age 24	repeated blows, careless treatment, beating twice a day	neither husband nor doctors spotted any external signs of maltreatment	a blind child, stayed several months in a hospital, the mother does not like her daughter the way she used to
B	age 2 years	male	repeated blows, fall on the floor, febrile convulsions	biological mother, minor brother: probably stepbrother	blows onto the face + repeated bites	doctor noticed bruises + bite marks on both arms	aggressive stepbrother, mother failed to respond, left the children alone together
C	2 months	female	physically and psychologically tormenting treatment—beating, strangling	biological mother, age 21, father: biological, age 30	reduction of air access by being wrapped in a blanket and subsequent beatings	a neighbour saw the abuse but did not report anything, the mother was involved in the violence	low social status
D	21 months	male	stepfather’s punches in the face	biological mother, non-biological father	extreme physical punishments for not being able to hold his stool	mother’s ignorance of the situation, child was afraid of his stepfather, a witness describes bruises	repeated hospitalisation, failure to thrive, alcohol was served to the child
E	13 months	female	father threw her on the bed and purulent meningitis	biological mother: biological father, age 16	repeated blunt violence of lesser intensity	child neglected, hypotrophic, psychomotor retardation, hospitalised in the ICU of the paediatric hospital	low social status, very young father
F	4 months	male	beating with palms and fists practically all over the body	biological mother, age 24, biological father	repeated direct action of blunt violence of low intensity	despite the child’s wheezing, a doctor was not called	mother was a prostitute—debts, reduced social and economic status, frequent quarrels, no emotional bond with the child
G	2 years	female	circumstantial evidence, without witnesses	biological mother, non-biological father	circumstantial evidence	no serious illness in the past	the biological father does not live with his family; impulsive or even explosive problem-solving techniques
H	5 years	female	the effects of intense violence caused by the mother	biological mother, biological father	old healed scars on the head, inappropriate conditions at home	delayed mental development of the child, child neglected, neither mother nor child had a doctor	unsatisfactory environment, dirt, clutter, repeated stays in infant institution
I	3 years	male	mother mentally failed to care for her son with a disability	biological mother, biological father	repeated random blows	husband did not notice signs of abuse	cerebral palsy
J	2 years	male	intense violence	biological mother, biological father	repeated violence, beatings, emaciation, and general decrepitation, without basic care, without the possibility of staying in fresh air	mental and physical abuse noticed by neighbours, repeated summoning of the mother to the police in the event of a fracture	infant care centre, mother does not like him

## Data Availability

Due to data sensitivity not applicable.

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
