# Peer review of "Early Identification of Risk of Child Abuse Fatalities: Possibilities and Limits of Prevention"

_children, 2022, doi:10.3390/children9050594_

Round 1

Reviewer 1 Report

Lines 192-195. The text is repeated on lines 206-7.

Line 241. …presented in tables. There is only one Table. It would be useful to put in a table the data of the registered 512 criminal offenses.

Line 343. Reference 24 is repeated.

Line 449. (comparison [24]). What does it mean?

Line 524. (mod 23.7). What does it mean?

Author Response

Lines 192-195. The text is repeated on lines 206-7.

Fully accepted

Line 241. …presented in tables. There is only one Table. It would be useful to put in a table the data of the registered 512 criminal offenses.

Fully accepted

Line 343. Reference 24 is repeated.

Fully accepted

Line 449. (comparison [24]). What does it mean?

Fully accepted

Line 524. (mod 23.7). What does it mean?

Fully accepted

Reviewer 2 Report

  • The objective of the study should be in the introduction section. should be changed to.
  • It is not clear whether the qualitative analysis was carried out on a sample of 10 children or on the 52 discussed at the beginning, obtained on the basis of a number of criteria.
  • What are the variables considered in the quantitative study?
  • In the title of the section called "Ethical aspects and limits of research" and according to the authors' comments, these are limitations of the study that should be included in the discussion section.
  • Has the study been evaluated by an ethics committee?
  • In the results section, 409 offences committed by known perpetrators are mentioned, but these issues are not addressed in the rest of the paper. The rest of the cases where the perpetrators are not known?
  • Why were the data studied collected during the years 2010, 2014 and 2019?
  • The study states that 52 autopsies were consulted, but data from only 10 autopsies were analysed according to qualitative analysis. Are these data correct?
  • The qualitative analysis of the autopsies is very basic, it is not known if they are categories, themes, although very sensitive testimonies are collected. A more complete analysis should be made and it should be discussed whether it is descriptive or interpretative.
  • The conclusions are very extensive. They need to be specified.
  • The conclusions cannot have bibliographical references and must respond to the objectives set
  • The references section needs to be updated, there are several that are very old

Author Response

The objective of the study should be in the introduction section. should be changed to.

Fully accepted

It is not clear whether the qualitative analysis was carried out on a sample of 10 children or on the 52 discussed at the beginning, obtained on the basis of a number of criteria.

Fully accepted, the information has been completed

What are the variables considered in the quantitative study?

Fully accepted, the text has been added

In the title of the section called "Ethical aspects and limits of research" and according to the authors' comments, these are limitations of the study that should be included in the discussion section.

I consider the chapter so important that I would like to keep it as a separate section, not as part of the discussion. However, if the reviewer really insists on moving it, I will move the text.

Has the study been evaluated by an ethics committee?

The text states that no live subjects were involved and that permission was obtained to collect the data. The ethics committee approval was for the broader context of the study within which this sub-study was conducted

In the results section, 409 offences committed by known perpetrators are mentioned, but these issues are not addressed in the rest of the paper. The rest of the cases where the perpetrators are not known?

Fully accepted, the information has been completed

Why were the data studied collected during the years 2010, 2014 and 2019?

Fully accepted, the information has been completed

The study states that 52 autopsies were consulted, but data from only 10 autopsies were analysed according to qualitative analysis. Are these data correct?

Fully accepted, the information has been completed, these data are correct

The qualitative analysis of the autopsies is very basic, it is not known if they are categories, themes, although very sensitive testimonies are collected. A more complete analysis should be made and it should be discussed whether it is descriptive or interpretative.

Fully accepted, the information has been completed, it was primary descriptive analysis, only the serial case study is interpretative

The conclusions are very extensive. They need to be specified.

Accepted, the conclusions were shortened, part of the text was moved to the discussion

The conclusions cannot have bibliographical references and must respond to the objectives set

Fully accepted

The references section needs to be updated, there are several that are very old

Partially accepted, I have added some current references we found and did not use; I believe some older resource could not be replaced because there are no newer and/or better (I made deep literature overview in this area).

Reviewer 3 Report

Thank you for your valuable research into this topic. There are significant and important learnings to be had by through this type of work.

However, the paper needs significant work. It is critical to set the scene for the reader as to what the context of the child protection/coronial/death review system is in the Czech Republic (I'm assuming this is the setting, but this is never explicitly stated). The methods section is confusing and difficult to follow and needs clarification. Although it appears that you have some interesting and important findings, the writing is unclear and results and discussion often mixed together - making it almost impossible to understand what the outcomes were and then how these findings could contribute to knowledge in this space. Some of this may be due to language issues so support from an English editor may help. 

This research could make a very valuable contribution to understanding the prevention of child maltreatment and associated deaths.

Author Response

It is critical to set the scene for the reader as to what the context of the child protection/coronial/death review system is in the Czech Republic (I'm assuming this is the setting, but this is never explicitly stated).

Accepted, the information has been completed

The methods section is confusing and difficult to follow and needs clarification.

Accepted, the chapter on methods has been revised and clarified

Although it appears that you have some interesting and important findings, the writing is unclear and results and discussion are often mixed together - making it almost impossible to understand what the outcomes were and then how these findings could contribute to knowledge in this space.

Partially accepted, the conclusions were shortened, part of the text was moved to the discussion. However, I'm not sure which part of the results mix with the discussion the reviewer is referring to. The first part reports on the quantitative analysis, the second on the qualitative analysis, which had both a descriptive and interpretive part. Does the reviewer wish to divide the results into subsections?

Some of this may be due to language issues so support from an English editor may help. 

Proofreading of the text was done by a native speaker, yet. Concerning extensive English revision, we would be happy to undergo it - please, instruct us about using one of the editing services

This research could make a very valuable contribution to understanding the prevention of child maltreatment and associated deaths.

Thank you very much for reading the text and for any valuable comments.